# The force-sensing peptide VemP employs extreme compaction and secondary structure formation to induce ribosomal stalling

Ting Su[1], Jingdong Cheng[1], Daniel Sohmen[1], Rickard Hedman[2], Otto Berninghausen[1], Gunnar von Heijne[2,3], Daniel N Wilson[1,4]*, Roland Beckmann[1]*

[1]Gene Center, Department of Biochemistry and Center for integrated Protein Science Munich, Ludwig Maximilian University of Munich, Munich, Germany; [2]Department of Biochemistry and Biophysics, Stockholm University, Stockholm, Sweden; [3]Science for Life Laboratory Stockholm University, Solna, Sweden; [4]Institute for Biochemistry and Molecular Biology, University of Hamburg, Hamburg, Germany

**Abstract** Interaction between the nascent polypeptide chain and the ribosomal exit tunnel can modulate the rate of translation and induce translational arrest to regulate expression of downstream genes. The ribosomal tunnel also provides a protected environment for initial protein folding events. Here, we present a 2.9 Å cryo-electron microscopy structure of a ribosome stalled during translation of the extremely compacted VemP nascent chain. The nascent chain forms two $\alpha$-helices connected by an $\alpha$-turn and a loop, enabling a total of 37 amino acids to be observed within the first 50–55 Å of the exit tunnel. The structure reveals how $\alpha$-helix formation directly within the peptidyltransferase center of the ribosome interferes with aminoacyl-tRNA accommodation, suggesting that during canonical translation, a major role of the exit tunnel is to prevent excessive secondary structure formation that can interfere with the peptidyltransferase activity of the ribosome.

*For correspondence: Daniel. Wilson@chemie.uni-hamburg.de (DNW); beckmann@genzentrum. lmu.de (RB)

**Competing interests:** The authors declare that no competing interests exist.

## Introduction

Many diverse nascent chain sequences have been identified that act as *cis*-regulatory polypeptides to promote ribosome stalling and thereby regulate gene expression. One of the most recently discovered examples is the Vibrio export monitoring polypeptide (VemP) that allows *Vibrio alginolyticus* to adapt to changes in salinity by modulating expression of components of the protein targeting machinery (*Ishii et al., 2015*). *V. alginolyticus* is a Gram-negative marine-estuarine bacterium that, unlike most bacteria, has two paralogs of SecDF (SecDF1 and SecDF2) to enhance protein export in cooperation with the SecYEG translocon. In Na$^+$-rich environments, *V. alginolyticus* utilizes the Na$^+$-dependent SecDF1; however, in Na$^+$-limiting environments *V. alginolyticus* switches to the Na$^+$-independent (presumably H$^+$-dependent) SecDF2 (*Ishii et al., 2015*). The switch between SecDF1 and SecDF2 is facilitated by the stalling activity of VemP. In Na$^+$-rich environments, VemP is targeted by its signal sequence (SS) to the SecDF1-SecYEG translocon (*Ishii et al., 2015*). The force applied to VemP during protein translocation prevents VemP-dependent ribosome stalling, thereby enabling the complete VemP to be translated and released from the ribosome. Under such conditions, SecDF2 expression is repressed due to sequestration of the ribosome-binding site (RBS) of *secDF2*

within a stem-loop structure in the mRNA (*Figure 1a*). In Na⁺-limiting environments, however, VemP is not translocated due to the inactivity of SecDF1, and the resulting lack of force on the VemP nascent chain leads to ribosomal stalling (*Figure 1a*). Ribosome stalling dissolves the stem-loop structure within the mRNA, exposes the RBS and thereby induces expression of Na⁺-independent SecDF2 (*Figure 1a*). In brief, the mechanisms by which VemP switches protein synthesis between the SecDF1 and SecDF2 paralogs is dependent on the application of a force to the signal sequence containing VemP nascent chain. This force serves as a direct indicator of the SecDF-SecYEG protein translocation activity, which in turn is dependent on the environmental conditions.

Biochemical studies have demonstrated that VemP-dependent stalling occurs when Q156 is in the P-site of the ribosome and that distinct amino acid residues within a 19aa window (138–156) of VemP are critical for efficient ribosome stalling (*Figure 1a*) (*Ishii et al., 2015*). Although VemP-

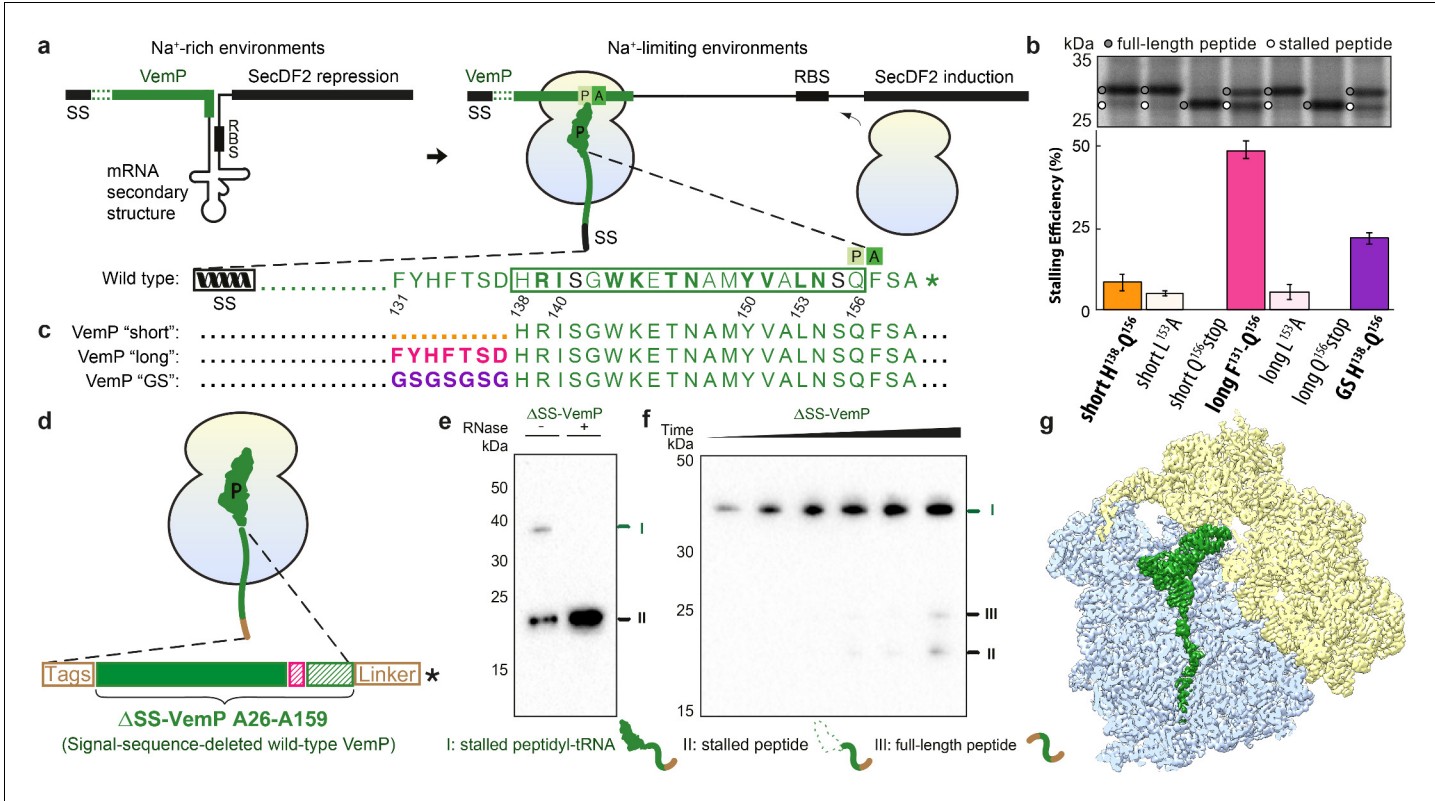

**Figure 1.** Biochemical and structural analysis of VemP stalling. (**a**) Schematic representation of the VemP-SecDF2 mRNA encoding VemP leader peptide with N-terminal signal sequence (SS) and C-terminal stalling region (green), followed by a stem-loop structure that sequesters the ribosome-binding site (RBS) of the SecDF2 gene (left). The translation arrest of VemP maintains the unfolded conformation of the mRNA allowing ribosome binding and induction of SecDF2 expression. The VemP stalling window H138–Q156 (boxed) is shown with critical (green bold) and important (green) residues highlighted (*Ishii et al., 2015*), and an asterisk indicating the stop codon (right). (**b–c**) In vivo pulse-chase analysis with different VemP constructs; VemP 'short' (H138–Q156, orange), VemP 'long' (F131–Q156, pink), VemP 'GS' (purple) and VemP mutants L153A and Q156*. (**d**) Schematic of the VemP-SRC used for cryo-EM. (**e–f**) Western blot against the N-terminal HA-tag of in vitro translated ΔSS-VemP detecting stalled peptidyl-tRNA (I), stalled free peptide (II) or full-length free peptide (III), (**e**) in the absence and presence of RNase A treatment, or (**f**) as a function of time (25, 40, 55, 70, 85, 100 min). (**g**) Transverse section of cryo-EM structure of the VemP-SRC showing the peptidyl-tRNA (green), with small and large subunits coloured in yellow and blue, respectively.

The following figure supplements are available for figure 1:

**Figure supplement 1.** Triplicates of the pulse-chase analysis.

**Figure supplement 2.** Classification of the VemP-SRC.

**Figure supplement 3.** Resolution of the VemP-SRC.

mediated regulation of SecDF2 expression is reminiscent of other peptide-stalling regulatory systems (*Ito and Chiba, 2013*; *Wilson et al., 2016*), such as SecM (*Nakatogawa and Ito, 2001*) and MifM (*Chiba et al., 2009*), the VemP stalling sequence and force-sensing mechanism appears to be completely distinct (*Ishii et al., 2015*; *Ito and Chiba, 2013*).

## Results

### Analysis of VemP sequence on efficiency of stalling

To ensure the stalling efficiency of the VemP constructs before initiating the structural analysis, we engineered three LepB-based constructs carrying C-terminal VemP residues, termed VemP 'short', 'long' and 'GS', respectively (*Figure 1b,c*). VemP 'short' carries the 19 amino acid segment that Ishii *et al* had chosen as the target for alanine scanning mutagenesis from VemP (138–156) (*Ishii et al., 2015*). VemP 'long' carries a 26 amino acid long segment including seven additional N-terminal residues of VemP (131–156), whereas VemP 'GS' is similar to the former two, except that residues 131–138 were substituted with a glycine-serine (GS) linker (*Figure 1b,c*). VemP 'short', 'long' and 'GS' were expressed in pulse-chase experiments in *E. coli* without any N-terminal signal sequence (see Material and methods). Using this assay, the major product observed to accumulate for VemP 'short' was full-length LepB-VemP fusion protein, whereas relatively little (10%) stalled peptide was detected (*Figure 1b,c* and *Figure 1—figure supplement 1*). In contrast, VemP 'long' containing the extended stalling window dramatically increased the stalling efficiency (*Figure 1b,c*). As expected, mutations previously reported to abrogate ribosome stalling (*Ishii et al., 2015*), L153A and Q156stop, led to a reduction in stalling efficiency for the VemP 'short' and particularly for the VemP 'long' constructs. VemP 'GS' also had a reduced stalling efficiency (*Figure 1b,c*). Taken together, these fusion proteins proved to be ineffective at inducing elongation arrest to the full extent in vivo. The influence of residues 131–138 on ribosome stalling and SecDF2 induction was not so extensively characterized in the previous study, where only mutation of residues 132 and 136–138 to alanine were reported to have little or no effect (*Ishii et al., 2015*). Thus, our results suggest that residues beyond the previously characterized 19-aa stalling window of VemP contribute to the efficiency of VemP-mediated translation arrest.

### Cryo-EM structure of the VemP-SRC

In order to investigate which residues of VemP are involved in establishing interactions with the exit tunnel during ribosome stalling, and in particular to decipher the molecular mechanism of force-sensitive ribosome stalling by VemP, we used cryo-EM to determine the structure of a VemP-stalled ribosome-nascent chain complex (VemP-SRC). Since we observed a contribution of the amino acids in VemP beyond the initially identified stalling window and to ensure the maximum ribosome stalling efficiency, the wildtype VemP sequence lacking only the signal sequence (residues 26–159, ΔSS-VemP), rather than a shorter stalling window, was used (*Figure 1d*). Translation of the ΔSS-VemP construct in an *E. coli* lysate-based translation system led to the accumulation of VemP-peptidyl-tRNA (about 40 kDa), of which the size was reduced to the expected free peptide (18 kDa) upon RNase treatment (*Figure 1e*). The robustness of stalling of the wildtype ΔSS-VemP was evident from the persistence of the VemP-peptidyl-tRNA, even with incubation times of up to 100 min where only minor fractions of full-length VemP were produced (*Figure 1f*). The VemP-SRC was purified using sucrose density gradient centrifugation and N-terminal affinity tags and subjected to cryo-EM analysis (see Materials and methods). *In silico* sorting of the VemP-SRC dataset yielded a major subpopulation of ribosomal particles that contained stoichiometric occupancy of the P-tRNA (*Figure 1—figure supplement 2*). Subsequent refinement resulted in a final reconstruction of the VemP-SRC (*Figure 1g*) with an average resolution of 2.9 Å and a local resolution of 2.5 Å within the core of the ribosome (*Figure 1—figure supplement 3* and *Table 1*). The electron density for the majority of the VemP nascent chain was well resolved within the exit tunnel, with local resolution ranging between 3.0 and 3.5 Å (*Figure 1—figure supplement 3*), thus enabling a molecular model for residues 120–156 of VemP to be built *de novo* (*Figure 2* and *Videos 1* and *2*).

**Table 1.** Refinement and Model Statistics.

| | |
|---|---|
| **Data collection** | |
| Particles | 400,024 |
| Pixel size (Å) | 1.084 |
| Defocus range (µm) | 1–2.5 |
| Voltage (kV) | 300 |
| Electron dose (e⁻/Å⁻²) | 25 |
| **Model Composition** | |
| Protein residues | 5615 |
| RNA bases | 4641 |
| **Refinement** | |
| Resolution (Å, 0.143 FSC) | 2.93 |
| Map sharpening B factor (Å$^2$) | -162.55 |
| FSC$_{Average}$ | 0.90 |
| **Validation** | |
| rmsd, bonds | 0.016 |
| rmsd, angles | 1.66 |
| Rotamers outliers (%) | 1.45 |
| Ramachandran outliers (%) | 1.00 |
| Ramachandran favoured (%) | 90.29 |
| Correct sugar puckers (%) | 98.77 |
| Good backbone conformations (%) | 81.74 |
| **Scores** | |
| MolProbity | 1.96 |
| Clash score, all atoms | 5.45 |

## Extreme compaction of the VemP nascent chain in the ribosomal tunnel

The VemP nascent chain adopts an extremely compacted conformation within the exit tunnel comprising two α-helices, located in the upper and lower regions of the tunnel, connected by an α-turn and loop (*Figure 2a–d* and *Video 1*). The α-helix located in the upper region of the tunnel is comprised of the C-terminal 10 residues (N147–Q156) of VemP and is directly linked to the P-tRNA and thereby located directly in the peptidyltransferase center (PTC) of the ribosome (*Figure 2a,b*). The α-helix located in the lower region of the tunnel is comprised of the N-terminal 10 residues (M120–S129) of VemP for which density is clearly visualized. We note that the N-terminal stretch of the VemP nascent chain from the lower α-helix to tunnel exit is less well resolved (*Figure 1—figure supplement 3*). The formation of α-helical secondary structure within the upper and lower regions of the exit tunnel is generally consistent with previous biochemical and structural studies (*Bhushan et al., 2010a*; *Kosolapov and Deutsch, 2009*; *Lu and Deutsch, 2005*; *Matheisl et al., 2015*; *Woolhead et al., 2004*). The linker between the two α-helices is located within the constriction of the exit tunnel, adjacent to ribosomal proteins uL4 and uL22 (*Figure 2a–d*). The linker is composed of 17 residues (130–146) that form two consecutive turns: one connecting the C-terminal α-helix which is a canonical α-turn (I140–K144), followed by a less well-ordered turn without hydrogen bonding (*Figure 2a–d* and *Video 2*). Protein folding events have been demonstrated to occur within the ribosomal exit tunnel (*Holtkamp et al., 2015*; *Lin et al., 2012*; *Marino et al., 2016*; *Nilsson et al., 2015*; *Tu et al., 2014*); (*Bhushan et al., 2010b*; *Matheisl et al., 2015*); Of those so far characterized, folding was shown to occur within the lower region of the tunnel, e.g. ADR1 (*Nilsson et al., 2015*) and within the vestibule at the periphery where the exit tunnel widens (*Cabrita et al., 2016*; *Nilsson et al., 2017*; *Trovato and O'Brien, 2016*; *Tu et al., 2014*).

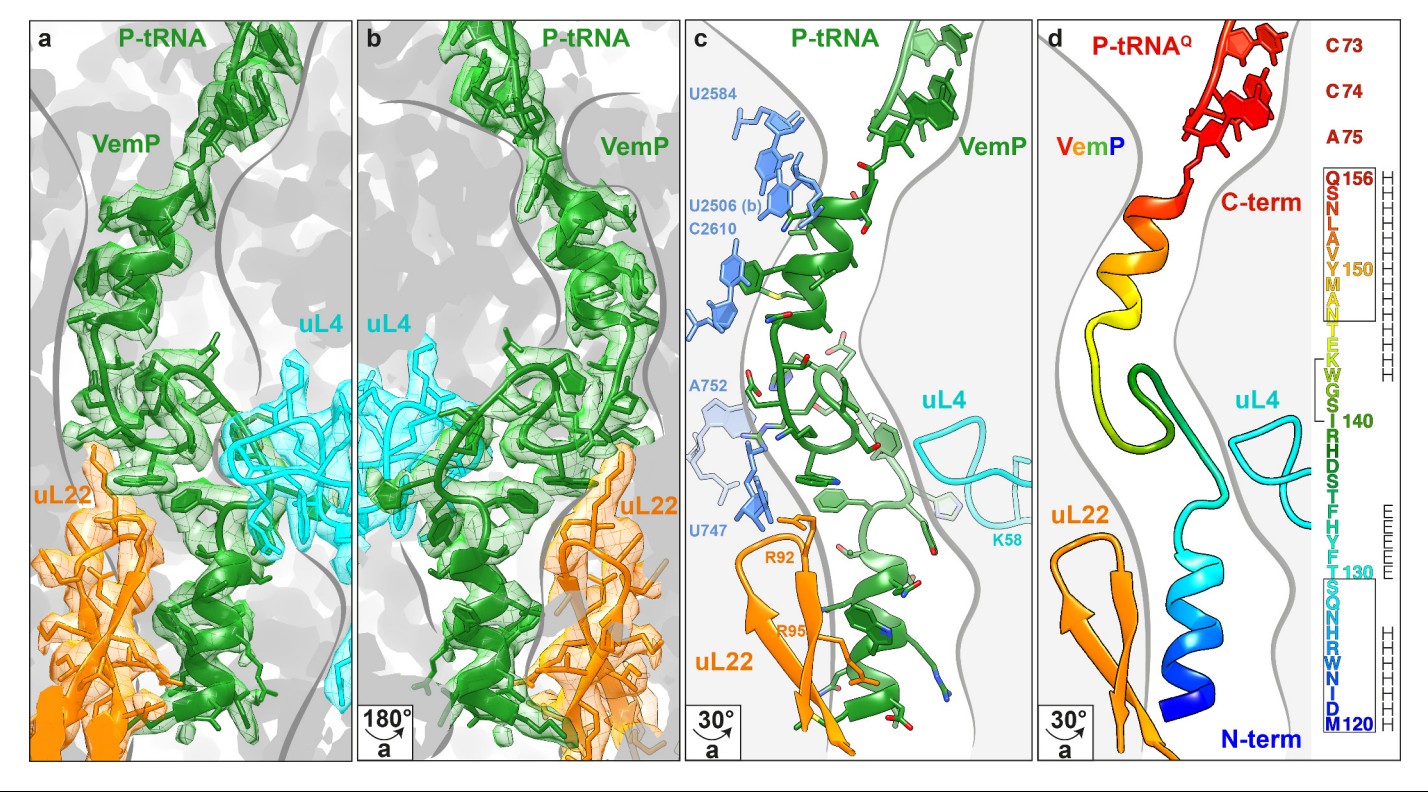

**Figure 2.** Overview of the VemP nascent chain in the ribosomal tunnel. (a–b) Isolated electron density and molecular model for the VemP nascent chain connected to the P-tRNA (green) in the ribosomal tunnel (grey) with ribosomal proteins uL4 (cyan) and uL22 (orange) highlighted. (b) is the same as (a) but rotated by 180°. (c) Model of the VemP nascent chain in the ribosomal tunnel highlighting ribosomal components that interact with VemP, namely, nucleotides of the 23S rRNA (blue) and ribosomal proteins uL4 (cyan) and uL22 (orange). (d) Rainbow representation of the VemP nascent chain with sequence coloured accordingly. The upper and lower α-helices of VemP are boxed and the α-turn in the linking loop region is bracketed. Secondary structure predictions for VemP are indicated with H (helix) and E (loop region).

Additionally, in the hCMV stalling peptide a C-terminal α-helix was also observed in the upper region of the tunnel in a similar yet slightly shifted position compared to the C-terminal helix of VemP (*Figure 3*) (*Bhushan et al., 2010b*; *Matheisl et al., 2015*). Structurally, the extensive compaction and secondary structure formation of VemP results in an unprecedented total of 37 residues being housed within the upper two thirds (approximately 50–55 Å) of the ribosomal exit tunnel, which contrasts with the 21–33 aa that were visualized within the exit tunnel for other stalling peptides, such as SecM (*Bhushan et al., 2011*; *Zhang et al., 2015*), MifM (*Sohmen et al., 2015*), TnaC (*Bischoff et al., 2014*; *Seidelt et al., 2009*) and CMV (*Bhushan et al., 2010b*; *Matheisl et al., 2015*) (*Figure 3*). When calculating the theoretical minimal number of residues for the VemP peptide chain to stretch all the way from the PTC to the tunnel exit, it would require at least

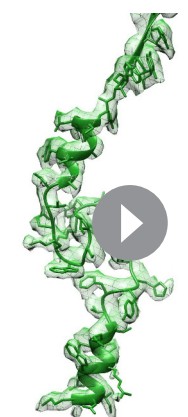

**Video 1.** Cryo-EM density and model for the VemP nascent chain. Video showing the quality of the cryo-EM density (green mesh) and fit of the molecular model (green ribbon) of the VemP nascent chain and CCA-end of the P-tRNA.

51 aa, which is in stark contrast to only 31 aa for MifM and 34 for SecM due to lack of compaction.

## Interaction of VemP within the ribosomal exit tunnel

Alanine scanning mutagenesis of VemP identified 10 residues that are critical for ribosome stalling, as well as an additional seven residues that significantly reduced stalling when mutated (*Ishii et al., 2015*). These residues encompass 17 of the 19-aa stalling window of VemP 'Short', with only the mutations of either S141A and S155A having no influence on ribosome stalling. Many of these residues establish specific interactions with components of the ribosomal exit tunnel, which is likely to explain their importance for ribosomal stalling (*Figure 4a–f*). For example, the sidechains of N154 and Y150, which are located in the upper C-terminal α-helix of VemP, can stack on the nucleobases of U2506 (*Figure 4a*) and C2610 (*Figure 4c*), respectively, of the 23S rRNA. Additionally, N154 comes within hydrogen bonding distance of U2584 (*Figure 4b*). Within the α-turn of VemP, K144 can hydrogen bond with the non-bridging phosphate-oxygen of U747 (*Figure 4d*), whereas W143 can stack upon the R92 sidechain located the tip of the β-hairpin of uL22 (*Figure 4e*). Consistently, alterations in uL22 have been shown to reduce the efficiency of VemP-mediated ribosome stalling (*Ishii et al., 2015*). The α-turn of VemP is additionally stabilized by stacking interactions between R139 of VemP and A752 of the 23S rRNA (*Figure 4f*). The seven residues (131–137) within the loop linking the α-turn of VemP to the lower N-terminal α-helix map within the VemP 'Long' region. As mentioned, alanine mutagenesis (Y132A, S136–H138A) suggested that the identity of some of these residues is not important for ribosome stalling, with the exception of the H138A mutation that had a modest effect on arrest efficiency (*Ishii et al., 2015*). Consistently, in the VemP-SRC structure, residues 135–138 of loop are less well-resolved and no sidechain interactions with the exit tunnel are observed. By contrast, the remaining loop residues (130–134) are better resolved. The sidechain of H133 comes within hydrogen bonding distance of the backbone of K58 of uL4 (*Figure 4g*), which may contribute to the improved stalling efficiency of the VemP 'Long' over the VemP 'Short' observed in *Figure 1b*. Additionally, the sidechain of W124, which is located in the lower N-terminal α-helix of VemP, stacks on the R95 sidechain of uL22 (*Figure 4h*). Collectively, the interactions observed in the VemP-SRC structure are consistent with previous biochemical and mutational analysis indicating that the majority of critical residues and interactions are located in the upper region of the tunnel, but that the additional residues within the lower region of the tunnel can also contribute to the efficiency of stalling.

## Silencing of the PTC by the C-terminal α-helix of VemP

Biochemical analysis indicates that VemP-stalled ribosomes arrest translation because peptide bond formation cannot occur between Gln156 of the VemP-peptidyl-tRNA in the P-site and Phe157 of the incoming aminoacyl-tRNA of the A-site (*Ishii et al., 2015*). Comparison of the conformation of nucleotides at the PTC of the VemP-stalled ribosomes revealed two nucleotides of the 23S rRNA, U2585 and U2506, which adopt conformations incompatible with peptide bond formation (*Figure 5a*). Accommodation of an aminoacyl-tRNA at the A-site of the PTC requires a transition of the PTC from an uninduced to an induced state, which involves a shift in the position of U2585 (*Figure 5b*) (*Schmeing et al., 2005*; *Youngman et al., 2004*). In the VemP-SRC, U2585 adopts a unique conformation, distinct from both the uninduced and induced U2585 conformations (*Figure 5b*). Moreover, the upper α-helix of VemP extends into the PTC and thereby directly prevents the transition of U2585 from the uninduced to the induced state, due to steric clashes with Asn154 of VemP (*Figure 5c*). In contrast to VemP, the hCMV helix does not directly prevent the analogous human nucleotide U4493 (*E. coli* U2585) from adopting the induced state. It is rather hCMV's penultimate Pro21, which is not part of the helix, that prevents this transition (*Figure 5d*). Additionally, U2506 is observed to adopt two distinct conformations, U2506(a) and U2506(b) in the VemP-SRC (*Figure 5a*). The U2506(a) conformation would sterically clash with the aminoacyl moiety of A-tRNA (*Figure 5e*) and is likely to contribute to prevent aminoacyl-tRNA accommodation at the A-site of the PTC. The induced state of the PTC also involves a shift in the position of U2506 (*Schmeing et al., 2005*; *Youngman et al., 2004*) (*Figure 5f*), which cannot occur in the VemP-SRC because the upper α-helix of VemP blocks the induced conformation of U2506 (*Figure 5g*), analogously to that observed for U2585 (*Figure 5c*). Unlike VemP, hCMV does not retain an uninduced conformation of human U4414 (*E. coli* U2506) but allows the induced conformation of U4414 since it

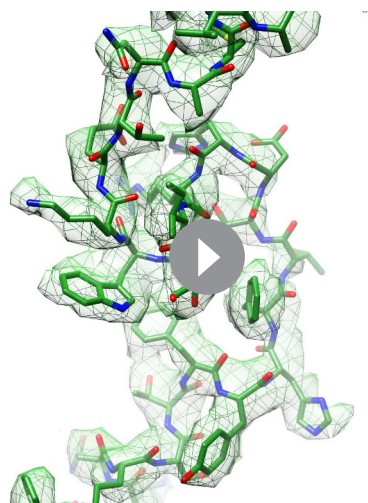

**Video 2.** Cryo-EM density and model for α-turn and loop of the VemP nascent chain. Video showing a zoom of the cryo-EM density (green mesh) and fit of the molecular model (stick representation) of the α-turn and loop of the VemP nascent chain.

is sterically less constrained by the more distant hCMV helix (*Figure 5h*) (*Matheisl et al., 2015*). Taken together, although α-helix formation has also been observed in the upper tunnel of an hCMV-stalled ribosome (*Matheisl et al., 2015*), the CMV α-helix did not directly extend into the PTC as seen here for VemP (*Figure 3f–g* and *Figure 5d,h*). Indeed, the two C-terminal proline residues of the CMV stalling peptide appeared to be more important for inactivating the PTC and inducing translation arrest (*Matheisl et al., 2015*). Notably, translation arrest via stabilization of the uninduced state has also been observed for other force-sensing stalling peptides on the ribosome, such as SecM (*Zhang et al., 2015*) and MifM (*Sohmen et al., 2015*). However, the mechanism is different since both, SecM and MifM, adopt entirely extended conformations within the tunnel, unlike VemP (*Figure 3*).

## Discussion

Collectively, the VemP-SRC structure reveals that VemP adopts an extremely compacted conformation within the ribosomal exit tunnel, comprising an upper and a lower α-helix connected by an α-turn and a loop. On the stalled ribosome, the upper α-helix of VemP encroaches directly upon the PTC and stabilizes an uninduced state of the PTC that prevents accommodation of the incoming A-tRNA and thereby promotes translation arrest (*Figure 6a*). These findings suggest that formation of secondary structure, such as an α-helix, directly at the PTC is likely to be detrimental for peptide bond formation and, therefore, for translation efficiency. Thus, the PTC of the ribosome and its immediate tunnel context must have evolved to generally disfavour excessive secondary structure formation. However, it also raises the question as to whether secondary structure formation at the PTC is used by the ribosome for specific nascent proteins to modulate the rate of translation, which could have important downstream consequences for co-translational protein folding and targeting events.

Relief of VemP-mediated ribosome stalling is proposed to result from the force exerted on the nascent chain during membrane insertion of the N-terminal signal sequence (*Ishii et al., 2015*). It is easy to imagine how, in the case of VemP, force application would inevitably prevent the formation of extensive secondary structure formation during translation, or at least lead to an unraveling of any secondary structure that does form within the tunnel, and thereby eventually allowing sterically the transition into the induced conformation of the PTC. As a result, peptide bond formation would be promoted again and translation elongation continued (*Figure 6b*). Interestingly, all three molecularly characterized force-sensing stallers employ the mechanism of preventing the adoption of the catalytically required induced state of the PTC by steric hindrance. This may be the favoured mechanism because the conformational trajectory from the uninduced to the induced state coincides with the direction of force applied to the nascent peptide (*Figure 6b*). Yet, it is surprising that the stalling peptides use completely different modes by which they interact with the ribosomal tunnel in a force-sensitive manner to achieve this goal.

## Materials and methods

### DNA manipulations

To test the effect of adding the extra C-terminal VemP sequence to the identified 19aa stalling window (*Ishii et al., 2015*), the VemP 'short' plasmid was constructed from a previously designed

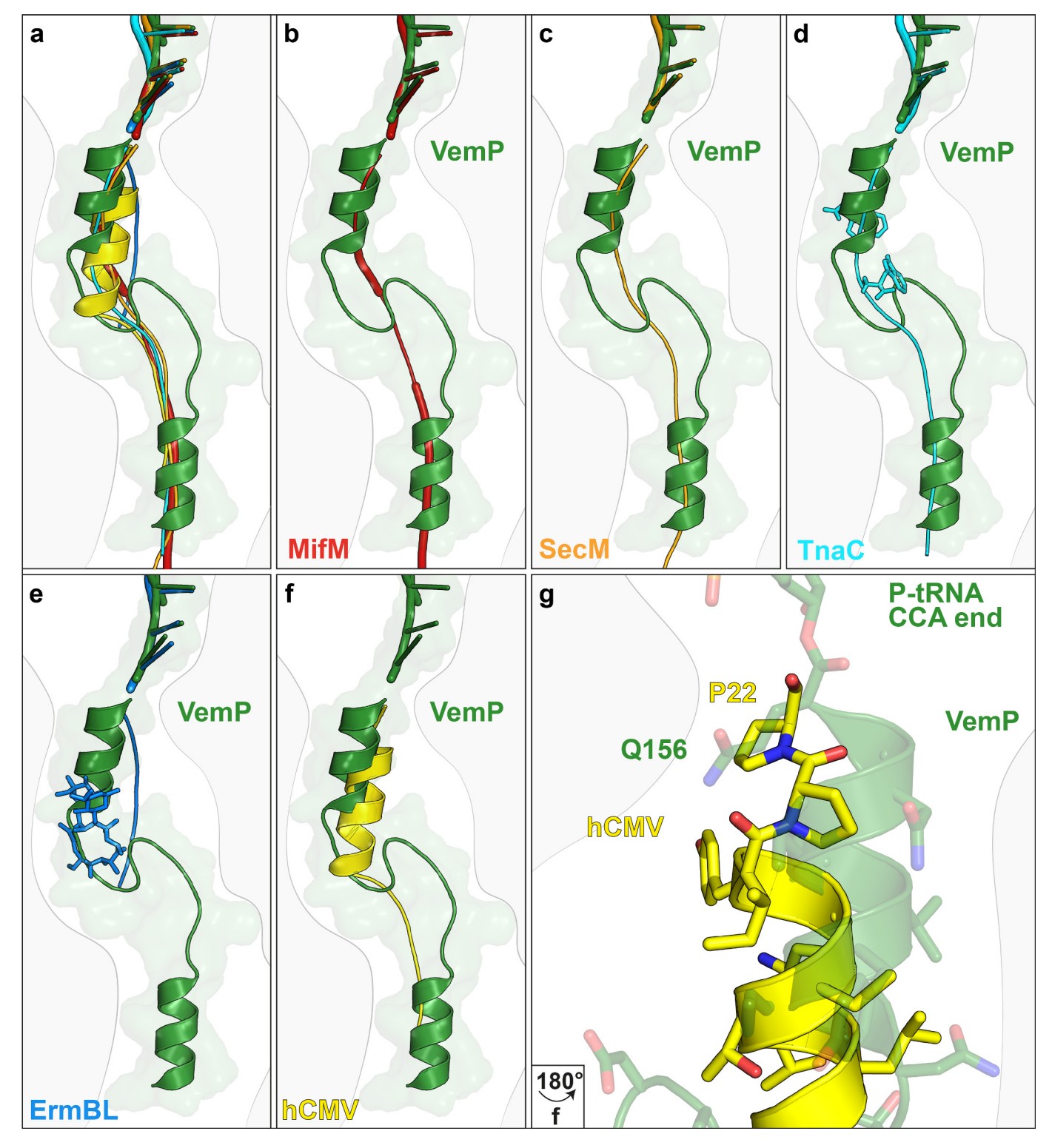

**Figure 3.** Comparison of the VemP nascent chain in the ribosomal tunnel with other stalling peptides. (a) Overall superposition of VemP (model in green, surface in light green) with MifM (red, PDB ID 3J9W) (*Sohmen et al., 2015*), SecM (orange, PDB ID 3JBU) (*Zhang et al., 2015*), hCMV (yellow, PDB ID 5A8L) (*Matheisl et al., 2015*), TnaC (cyan, PDB ID 4UY8) (*Bischoff et al., 2014*) and ErmBL (blue, PDB ID 5JU8)(*Arenz et al., 2016*) in the ribosomal tunnel (light grey). The CCA-end of the P-tRNA is shown for reference. The ribosomal tunnel is shown schematically in grey. (b–f) Superposition of VemP (green) with each arrest peptide described above. For (d) TnaC and (e) ErmBL, the two tryptophans (cyan) and the one
*Figure 3 continued on next page*

Figure 3 continued

erythromycin (blue) molecule are shown, respectively. (**g**) A zoomed-in view of (**f**) showing the superposition of VemP (green) and hCMV (yellow) in the C-terminal helical part including the side chains. The last residue, Q156 in case of VemP and P22 in case of hCMV, connecting to the CCA end of tRNA is labelled.

pING1 plasmid carrying a *lepB*-based construct harboring a SecM arrest peptide (*Ismail et al., 2012*). In order to produce a soluble, non-membrane targeted variant, transmembrane helix 1 and 2 of Lep were deleted using PCR. To replace the SecM arrest peptide with the VemP arrest sequence, the plasmid was amplified using a primer pair producing a linear plasmid lacking the SecM arrest peptide. Gibson assembly (*Gibson et al., 2009*) was then used to ligate synthesized oligonucleotides corresponding to the VemP arrest peptide (in bold) and its following three residues, **HRISGWKE TNAMYVALNSQ**FSA, into the plasmid. Overlap PCR (*Liu and Naismith, 2008*) was used to replace the seven Lep residues just upstream of the VemP 'short' arrest peptide with the corresponding seven residues from the native VemP and by GSGSGSG to generate VemP 'long', FYHFTSDHRISG WKETNAMYVALNSQFSA, and VemP 'GS', GSGSGSGHRISGWKETNAMYVALNSQFSA, respectively. Overlap PCR was also used to generate other variants of VemP used in pulse-labelling analysis.

For in vitro translation and cryo-grid sample preparation, the full VemP gene without the signal sequence (ΔSS-VemP, A26-A159) was cloned from synthesized *V. alginolyticus* genome by KOD XtremeTM Hot-Start DNA Polymerase (Novagen, MA, USA), and subsequently inserted to modified plasmid p7XNH3 by FX cloning method (*Geertsma, 2014*). The complete construct contains an N-terminal His-tag for purification, a FLAG-tag for detection and a TEV-protease cleavage site as a linker sequence. The C-terminus following the ΔSS-VemP comprises a spacer containing a 3C-protease cleavage site and an HA-tag.

MPWIYLRKLILLLFAMVLLPVHVSAAQIDHKAHVPHFSKLQPFVAVSVSPNSSVDFSEASEESSQSPVSE-GHASLDSVALFNSQRWTSYLREGLDDEHVDFVGDLTTPFYADAGYAYSLMDINWRHNQSTFYHFTSDHRI SGWKETNAMYVALNSQFSA(TAA-Stop x2) is the total amino acid sequence of the construct.

## *In vivo* pulse-labelling

*E. coli* MC1061 carrying plasmids with the different *lepB*-based constructs described in the paper were cultured overnight at 37°C in M9 minimal medium supplemented with 19 natural amino acids (1 µg/ml; no methionine, 100 µg/ml thiamine, 0.1 mM $CaCl_2$, 2 mM $MgSO_4$, 0.4% (w/v) fructose, and 100 µg/ml ampicillin). Overnight cultures were back-diluted to $OD_{600}$ ~0.05–0.1 and grown 2.5 to 3.5 hr to an $OD_{600}$ of 0.2–0.35. Protein expression was induced with 0.2% (w/v) arabinose for 5 min, whereafter expressed proteins were radiolabelled with [$^{35}$S]methionine for 2 min at 37°C. Ice-cold trichloroacetic acid (TCA) was added to a final concentration of 10% and the samples were incubated on ice for 30 min. The samples were centrifuged for 5 min at 20,800 x g at 4°C and the precipitates were washed with cold acetone, and spun again for 5 min at 20,800 x g at 4°C. The precipitates were resolubilized in Tris-SDS solution (10 mM Tris-HCl, pH 7.5, and 2% SDS) at 95°C for 10 min and spun for 5 min at room temperature. The protein of interest was then immunoprecipitated using anti-LepB antibody and the resulting samples were prepared for SDS-PAGE analysis with Laemmli sample buffer supplemented with 400 µg/ml RNase A. Gels were analyzed with a Fuji FLA-3000 phosphorimager and ImageGauge V4.23 software. Quantification of protein bands was performed using EasyQuant (in-house developed software). The fraction of full-length values (fFL) was calculated using the formula $fFL = IFL / (IFL+IA)$, where IFL is the quantified intensity of the full-length protein band and IA is the quantified intensity of the arrested protein band. Experiments were repeated in triplicate and standard errors of the mean were calculated.

## *In vitro* transcription and translation

For the RNase treatment assay and the final large scale purification for cryo-grid, RTS 100 *E.coli* HY Kit (5 PRIME) was used (transcription and translation coupled). 500 µL reaction was incubated at 30°C for 35 min with 12.6 pmol PCR-product template generated upon the ΔSS-VemP construct. RNase was added to the reaction where indicated, and incubated at 30°C for an additional 10 min. For the time course, the same amount of PCR-product template was used with the PURExpress In Vitro Protein Synthesis Kit (New England Biolabs #E6800S, transcription and translation coupled).

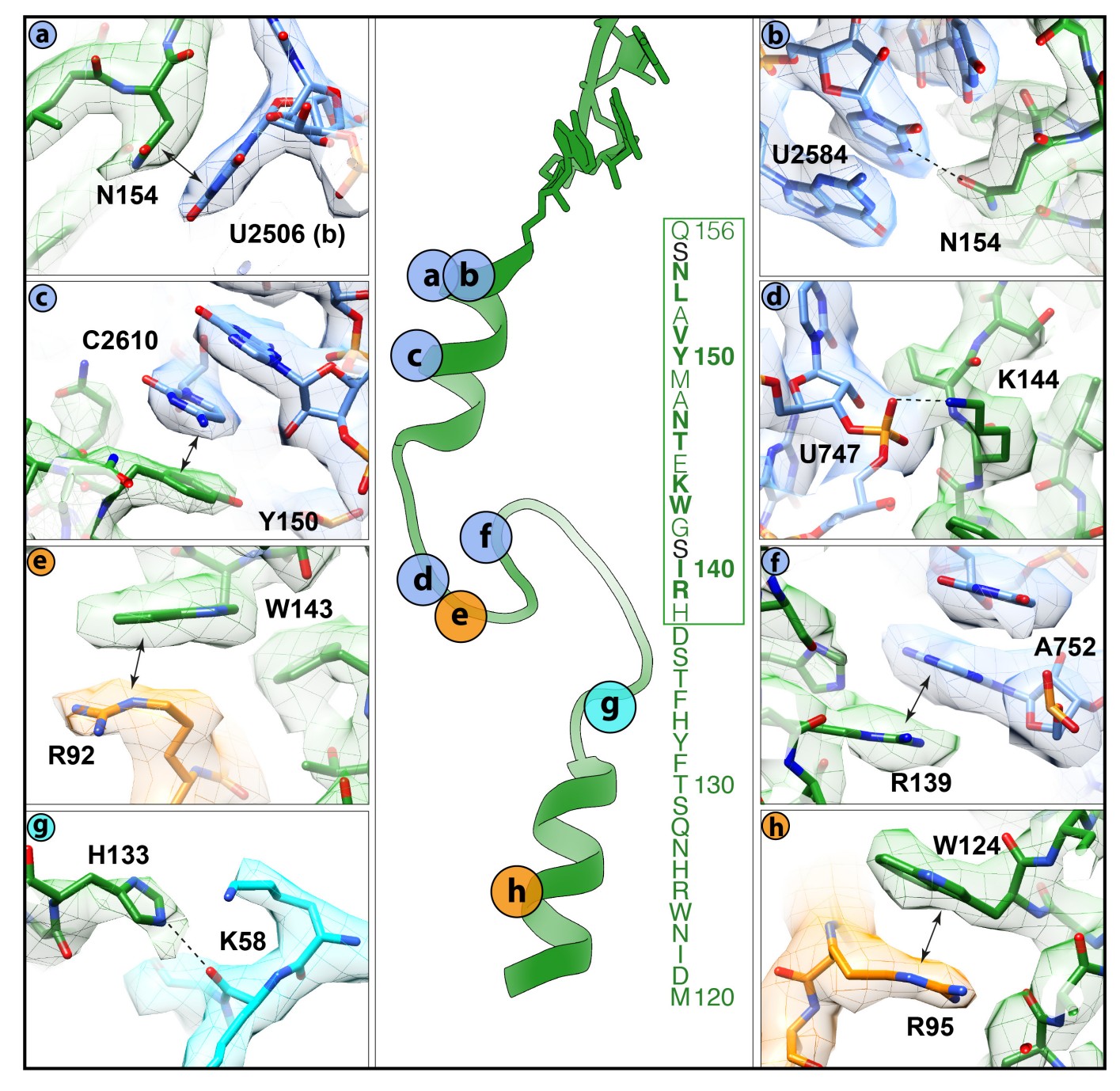

**Figure 4.** Interactions of the VemP nascent chain with the ribosomal tunnel (with density shown). (**a**) N154 of VemP (green) stacks upon U2506(b) of the 23S rRNA (blue). (**b**) N154 of VemP is within hydrogen bond distance of nucleotide U2584. (**c**) Y150 of VemP stacks upon C2610. (**d**) K144 of VemP forms a salt bridge with U747. (**e**) W143 of VemP stacks with R92 of uL22 (orange). (**f**) R139 of VemP stacks upon A752. (**g**) W124 of VemP stacks upon R95 of uL22. (**h**) H133 of VemP interacts with K58 of uL4 (cyan). The middle panel maps the approximate locations of the interactions and includes the sequence of VemP with critical (bold) and important (boxed green) residues highlighted (*Ishii et al., 2015*). In panels (**a–h**) the electron density for the VemP-SRC map (mesh) is coloured according to either VemP (green), 23S rRNA (blue), uL4 (cyan) or uL22 (orange).

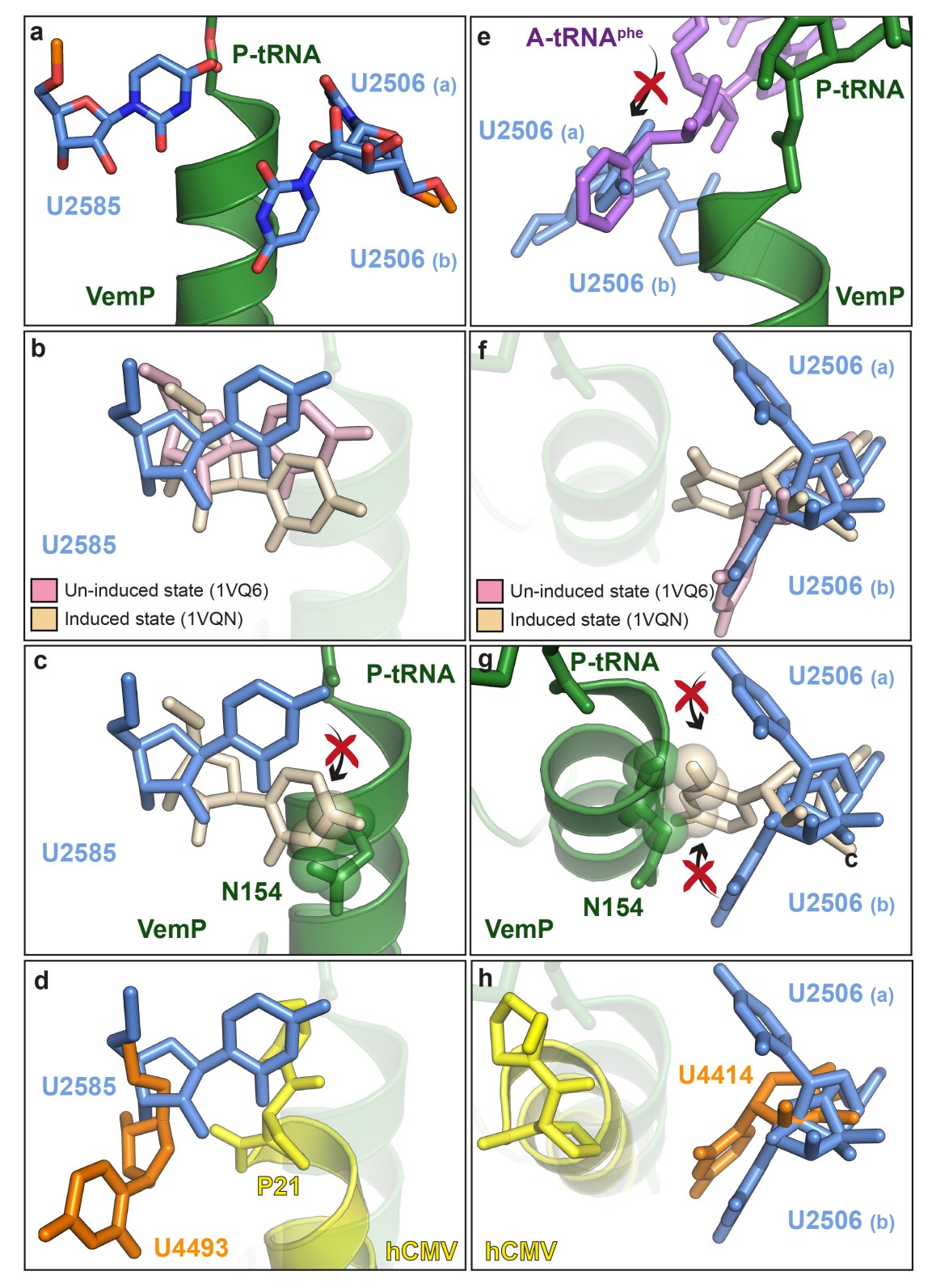

**Figure 5.** VemP stabilizes the uninduced state of the PTC to inhibit A-tRNA accommodation. (**a**) Conformation of U2585 and U2506 relative to VemP (green) at the PTC of the VemP-SRC. (**b**) Conformation of U2585 (blue) in the VemP-SRC compared with the uninduced (pink, PDB ID 1VQ6) and induced (tan, PDB ID 1VQN) states (*Schmeing et al., 2005*). (**c**) N154 of the upper α-helix of VemP clashes (indicated by spheres) with the induced conformation of U2585 (*Schmeing et al., 2005*). (**d**) Conformation of human U4493 (orange, numbering according to PDB ID 5A8L and 5AJ0, being

*Figure 5 continued on next page*

 Research article

Biochemistry | Biophysics and Structural Biology

**Figure 5 continued**

equivalent to U4531 in PDB ID 4UG0 and 4V6X) in the hCMV-SRC compared with *E. coli* U2585 (blue) in the VemP-SRC. (e) The U2506(a) conformation overlaps the binding position of an accommodated Phe-tRNA (purple) at the A-site of the PTC (*Schmeing et al., 2005*). (f) Conformation of U2506 in the VemP-SRC (blue) compared with the uninduced (pink) and induced (tan) states (*Schmeing et al., 2005*). (g) N154 of the upper α-helix of VemP clashes (indicated by spheres) with the induced conformation of U2506 (*Schmeing et al., 2005*). (h) Conformation of human U4414 (orange, numbering according to PDB ID 5A8L and 5AJ0, being equivalent to U4452 in PDB ID 4UG0 and 4V6X) in the hCMV-SRC compared with U2506 (blue) in the VemP-SRC. hCMV stalling peptide is shown in yellow in (d) and (h).

Reactions were incubated at 37°C for various times (25/40/55/70/85/100 min). For western blotting, reaction products were separated by either home-made or commercial NuPAGE 12% Bis-Tris gels (Invitrogen, CA, USA) with 1x MOPS buffer. Proteins were blotted to nitrocellulose membrane (Carl Roth, Germany), incubated with mouse anti FLAG M2 HRP (Sigma, Germany) and visualized by ChemiDoc MP System (Bio-Rad).

## Purification of the VemP-SRC

500 µL in vitro translation reaction was loaded onto 10–50% sucrose gradient prepared with Buffer C (25 mM pH 7.2 HEPES-KOH, 100 mM KOAc, 10 mM Mg(OAc)$_2$, 0.01% DDM, 1/1,000 complete protease inhibitor (Roche, Germany), 0.2 U/mL RNase, 2 mM 2-mercaptoethanol) and centrifuged for 3 hr in a Beckman coulter SW40 Ti swinging bucket rotor with 35,000 r.p.m. at 4°C. Gradients were separated on a Biocomp Gradient Station and fractions containing 70S ribosomal particles were collected and loaded onto a Talon metal affinity chromatography column (1.5 ml resin) pre-equilibrated in 10 mL buffer C containing 10 µg/mL bulk tRNA. The column was washed with 25 ml buffer C until no significant absorption (OD$_{260}$) could be detected in the wash fractions. The VemP-SRC, bound to the Talon matrix by the VemP N-terminal His-tag, was eluted in 750 µL buffer C containing 150 mM imidazole. The elution was pelleted for 4 hr 20 min in a Beckman Coulter TLA 120.2 fixed-angle rotor with 51,000 r.p.m. at 4°C. 15.6 pmol VemP-SRC pellet was resuspended in ice-cold buffer C without DDM, aliquoted and snap-frozen.

## Cryo-electron microscopy and single particle reconstruction

*E. coli* VemP-SRC (6 OD$_{260}$/mL) was applied to 2 nm pre-coated Quantifoil R3/3 holey carbon supported grids and vitrified using Vitrobot Mark IV (FEI). Data collection was performed using EM-TOOLS (TVIPS GmbH) on a Titan Krios transmission electron microscope equipped with a Falcon II direct electron detector (FEI) at 300kV at a pixel size of 1.084 Å and a defocus range of 1–2.5 µm. 10 frames (dose per frame of 2.5 e$^-$/Å$^2$) were aligned using MotionCor2 (*Li et al., 2013*). Power-spectra, defocus values and astigmatism were determined with CTFFIND4 software (*Rohou and Grigorieff, 2015*). 5735 micrographs were manually inspected in real space and in the meantime filtered by threshold of resolution at 4.5 Å and astigmatism at 5% resulting in 4849 micrographs. 850,433 particles were automatically picked by Gautomatch (http://www.mrc-lmb.cam.ac.uk/kzhang/) and single particles were processed using RELION 1.4 (*Scheres, 2012*). After 2D classification, 789,006 particles were subjected to 3D refinement using *E. coli* 70S ribosome as reference structure, followed by several further rounds of 3D classifications, including tRNA-focused sorting (64 rounds) and sorting (58 rounds) with a ribosome mask (*Figure 1—figure supplement 2*). One major class containing 400,024 particles (62%) was further refined, resulting in a final reconstruction with an average resolution of 2.9 Å (0.143 FSC) (*Figure 1—figure supplement 3*). The map was subsequently B-factored by EMBfactor (*Fernández et al., 2008*) with the FSC. Finally, the local resolution was calculated using ResMap (*Kucukelbir et al., 2014*).

## Molecular modeling and refinement of the VemP-SRC

The molecular model for the ribosomal proteins and rRNA of the 70S ribosome of the VemP-SRC was based on the molecular model from cryo-EM reconstruction of the *E. coli* 70S ribosome (PDB ID 5JU8) (*Arenz et al., 2016*), except that the bL31 was based on the (PDB ID 5LZD) (*Fischer et al., 2016*). The molecular model for Gln-tRNA was based on a crystal structure (PDB ID 1GSG) (*Rould et al., 1989*). The molecular models were initially fitted as a rigid body into the cryo-EM density map of the corresponding stalled complex using UCSF Chimera (*Pettersen et al., 2004*). The

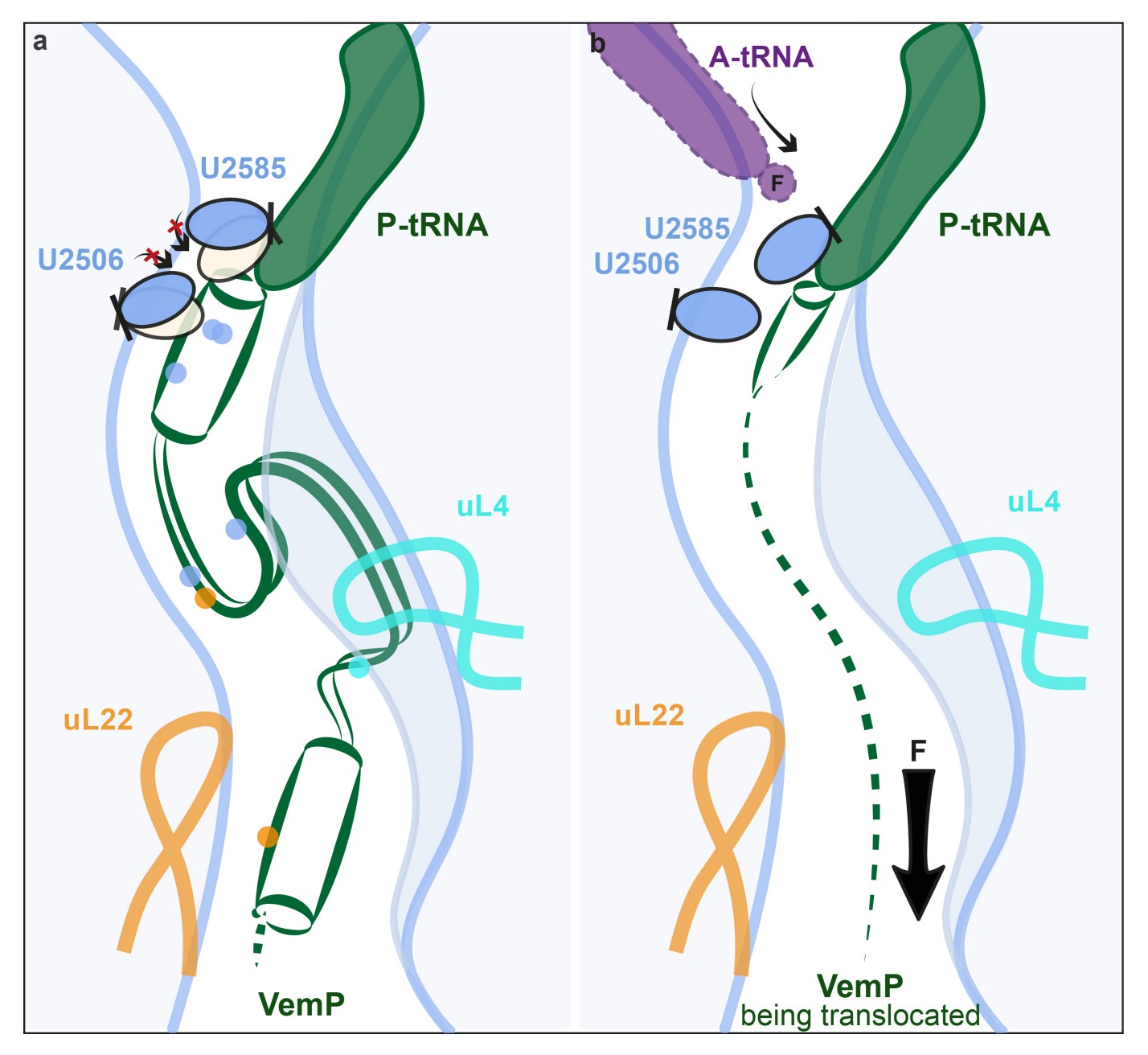

**Figure 6.** Model for VemP-mediated translation stalling and relief. (a–b) Schematic summarizing the molecular basis for (a) VemP-dependent translation arrest and (b) relief of arrest via the pulling-force (indicated by arrow) on VemP protein translocation.

complete atomic model of the VemP-SRC was manually adjusted using Coot (*Emsley and Cowtan, 2004*) and refined using phenix.real_space_refine with restraints obtained by phenix.secondary_structure_restraints (*Adams et al., 2010*). The Phenix refined model was further refined using REFMAC (*Vagin et al., 2004*) to validate the overfitting as previously described (*Brown et al., 2015*) (*Figure 1—figure supplement 3*). The statistics of the refined model were calculated using Molprobity (*Chen et al., 2010*) are presented in *Table 1*.

## Figure preparation

The protein secondary structure prediction of the native VemP sequence was performed using the MPI bioinformatics Toolkit (*Alva et al., 2016*) with the prediction method written by B. Rost (*Rost, 2001*). Figures showing electron densities and atomic models were generated using either UCSF Chimera (*Pettersen et al., 2004*) or PyMol Molecular Graphics Systems (Version 1.8 Schrödinger, LLC,).

## Accession codes

The cryo-electron microscopy map for the VemP-SRC has been deposited in the EMDataBank with the accession code EMD-3713. The respective coordinates for electron-microscopy-based model of the VemP-SRC are deposited in the ProteinDataBank (5NWY).

## Acknowledgements

We thank our technical team, especially Charlotte Ungewickell, and IT support, particularly André Heuer and Christian Schmidt. This research was supported by grants from the Deutsche Forschungsgemeinschaft GRK 1721 and FOR1805 (to RB and DNW), a DFG fellowship through the Graduate School of Quantitative Biosciences Munich (QBM, to TS) and the European Research Council (Advanced Grant CRYOTRANSLATION to RB), and the Swedish Cancer Foundation, the Swedish Research Council and the Knut and Alice Wallenberg Foundation (to GvH).

## Additional information

### Funding

| Funder | Grant reference number | Author |
| --- | --- | --- |
| Graduate School of Quantitative Biosciences Munich | Graduate Student Fellowship | Ting Su |
| Cancerfonden | | Gunnar von Heijne |
| Vetenskapsrådet | | Gunnar von Heijne |
| Knut och Alice Wallenbergs Stiftelse | | Gunnar von Heijne |
| Deutsche Forschungsgemeinschaft | FOR1805 | Daniel N Wilson |
| Deutsche Forschungsgemeinschaft | FOR1805 | Roland Beckmann |
| European Research Council | | Roland Beckmann |
| Deutsche Forschungsgemeinschaft | GRK1721 | Roland Beckmann |

The funders had no role in study design, data collection and interpretation, or the decision to submit the work for publication.

### Author contributions

TS, Writing—original draft, Writing—review and editing, Sample preparation, Cryo-EM data processing, model building and refinement; JC, Cryo-EM model building and refinement; DS, Helped with sample preparation, cryo-EM data processing and model building; RH, Pulse-labeling analysis; OB, cryo-EM data collection; GvH, Supervision, Writing—original draft, Writing—review and editing; DNW, Conceptualization, Supervision, Validation, Writing—original draft, Project administration, Writing—review and editing; RB, Conceptualization, Supervision, Funding acquisition, Validation, Writing—original draft, Project administration, Writing—review and editing

### Author ORCIDs

Ting Su, http://orcid.org/0000-0002-3185-8144
Jingdong Cheng, http://orcid.org/0000-0003-4442-377X

Gunnar von Heijne,  http://orcid.org/0000-0002-4490-8569
Daniel N Wilson,  http://orcid.org/0000-0003-3816-3828
Roland Beckmann,  http://orcid.org/0000-0003-4291-3898

## Additional files

### Major datasets

The following dataset was generated:

| Author(s) | Year | Dataset title | Dataset URL | Database, license, and accessibility information |
|---|---|---|---|---|
| Su T, Cheng J, Sohmen D, Hedman R, Beringhausen O, von Heijne G, Wilson DN, Beckmann R | 2017 | 2.9 A cryo-EM structure of VemP-stalled ribosome-nascent chain complex | http://www.rcsb.org/pdb/explore/explore.do?structureId=5NWY | Publicly available at the RCSBProtein Data Bank (accession no: 5NWY) |

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
