## [Decision Letter]

Thank you for submitting your article "The force-sensing peptide VemP employs extreme compaction and secondary structure formation to induce ribosomal stalling" for consideration by *eLife*. Your article has been reviewed by three peer reviewers, one of whom, Rachel Green (Reviewer #1) is a member of our board of Reviewing Editors and the evaluation has been overseen John Kuriyan as the Senior Editor. The following individuals involved in review of your submission have agreed to reveal their identity: Koreaki Ito (Reviewer #3).

The reviewers have discussed the reviews with one another and the Reviewing Editor has drafted this decision to help you prepare a revised submission.

We have received comments from three expert reviewers all of whom find the structure of the VemP-dependent stalling peptide complexed with the ribosome to be of high quality and broad general interest. The reviewers raise several issues, a few minor computational questions, and some concerns with the biochemical experiments presented in Figure 1 (and whether they should be included in the body of the manuscript given their limited utility), and a request for a clearer comparison with CMV-stalling peptide, and the possible inclusion of an additional citation. The authors should address the specific issues raised by the reviewers. Otherwise, once these minor issues are addressed, the manuscript should be acceptable for publication at *eLife*.

Reviewer #1:

This manuscript by Su et al. described a cryoEM structure of *E. coli* ribosomes stalled on an mRNA encoding the VemP peptide and characterized in some molecular detail. Here, the authors redefine the critical features of the peptide sequence needed for efficient stalling (extending the amino acid window by 7 aa to 26 in total) and then isolate from in vitro translation reactions the stalled ribosome complex. Elimination of the signal sequence from the peptide is critical in vivo for allowing stalling to occur but is apparently not necessary in the PURE system that is being utilized in vitro. The authors then present a well-resolved structure of the ribosome, with high resolution for the ribosome as a whole (2.9 Å), and with only slightly reduced resolution in the exit tunnel (3-3.5 Å). With this resolution, the authors are able to identify the overall highly compacted structure of the nascent peptide (which is quite distinct from that observed for other force-regulated stalling sequences like MifM and SecM), the details of interactions of the peptide with the tunnel that specify the stall, and an "uninduced" PTC that explains why catalysis does not proceed (A site tRNA is sterically hindered from binding).

Overall, the data are clean and convincing, and the structure provides insight into a different way in which a nascent peptide in the ribosome exit tunnel can regulate protein synthesis. This report will be of interest to readers of *eLife* and I recommend publication.

Points:

1) The term "translocation" used in the manuscript refers to protein translocation, yet to someone in the translation (not secretion) field, this refers to the reaction catalyzed by EFG. This could be clarified throughout the manuscript.

2) Discussion of Na^+^ regulation was a bit hard to integrate into the results of the manuscript. It might be worth clarifying that the results presented do not shed light on this particular subject but that this was how the motif was originally identified/studied.

3) More detailed comparison of the CMV stalling peptide and VemP could be made. While Figure 3 clearly shows that there is considerably more compaction in VemP peptide than in MifM or SecM, the distinction relative to the CMV peptide is less obvious. This should also be addressed in the text.

4) The induced fit of the ribosome PTC was documented structurally by Schmeing et al., 2005 and the results here nicely agree. However, earlier biochemical work (Youngman et al., 2004) detailed an induced fit mechanism based on differences in reactivity of intact tRNAs and minimal A site substrates (like puromycin). It might be worth considering whether this study should also be cited.

Reviewer #2:

Su et al., present an interesting structural analysis of the regulation of SecDF2 translation mediated by VemP-dependent stalling of the 70S ribosome. They obtained a cryo-EM reconstruction of the stalled and affinity-purified 70S complex. They reached the very high resolution of 2.9 Å and the high quality of the map facilitated de-novo modelling of the nascent VemP peptide located in the ribosomal exit tunnel. The structural analysis reveals an unprecedented, highly compacted state of the nascent peptide with two α-helices formed. Importantly, specific interactions of the VemP peptide with ribosomal residues of the tunnel and a remodeling of critical bases of the ribosomal peptidyl-transferase center explains the molecular mechanism of VemP-dependent stalling of translation. Overall, the data are of very high quality and the paper is very well written.

Reviewer #3:

Here is another contribution from the Beckmann-Wilson groups reporting the structures of ribosome-nascent polypeptide complexes, which have stalled after translating special amino acid sequences called arrest sequences. This time, the authors chose VemP, a regulatory nascent polypeptide from *Vibrio alginolyticus*, which was reported recently to play a central role in the adaptation of this marine bacterium to environments of low sodium concentrations. VemP is expressed from the upstream open reading frame in front of secDF2, encoding a proton-utilizing export enhancer, and undergoes translational elongation arrest, which enables the expression of secDF2 by disrupting the secondary structure of mRNA and exposing the ribosome-binding sequence of secDF2. VemP is a "Sec-monitoring substrate," and its elongation arrest is cancelled in a manner coupled with its translocation across the membrane. When this species of bacteria are living in seawater, they express only SecDF1, which can only use a gradient of Na^+^. Accordingly, the cellular Sec activity declines when they encounter a low salinity environment. The Sec dysfunction acts to prolong the stalling of the VemP-translating ribosome, which in turn enhances translation of SecDF2 by the ribosomal pool in the cell. VemP is the third member of the force-sensitive class of regulatory arrest peptides, following SecM and MifM in the order of discovery.

The study has been conducted with a high standard of design and skills, and the results provide additional insights into an emerging concept in molecular biology: nascent chain handling of the translating ribosome. Taken together with the results of similar analyses carried out to date, the present study reveals the high degree of diversity, with which different nascent polypeptides interact with the ribosomal components. Notably, the nascent chain of VemP establishes extensive contacts with the ribosome and undergoes an unprecedented extent of secondary structure formation in the ribosomal interior. I recommend publication of this work as it presents a clear case of nascent chain-instructed ribosomal dysfunction and force-dependent translation recovery.

Specific comments.

1. Initially, I did not understand exactly what were the VemP constructs that the authors referred to "short" and "long". Inspection of Materials and Methods then made me realize that they were not really VemP constructs; rather, the authors inserted segments of VemP comprising the 19 and the 26 amino acids into the LepB sequence. The authors must have expected that these amino acid sequences would fully halt translational elongation when transplanted into a sequence context of an unrelated protein, LepB in this case. However, Ishii et al. only determined the C-terminal end of the VemP arrest sequence, leaving the N-terminal boundary obscure. Moreover, it is known that the arrest proficiency can be affected by the nature of the N-terminally adjacent sequences, although the LepB sequence may be expected to be inert in this respect. The results in Figure 1 show that neither the short nor the long construct exhibited full elongation arrest in vivo; less than 10% and ~50% efficiency, respectively, for "short" and "long". By contrast, the full-length delta-signal VemP gives nearly 100% arrest in vivo as shown by Ishii et al and in vitro as shown in Figure 1. In my opinion, the information content of Figure 1 is quite limited. Most importantly, the authors anyway used the full-length delta-signal VemP to prepare the samples of ribosome-nascent chain complexes for cryo-EM analysis. These considerations lead me to suggest that the clarity of the paper will be improved if the authors remove Figure 1 as well as the schemes of the short, the long and the GS constructs, from the main body of the manuscript. A simple description something like the following might be sufficient and appropriate: "We constructed LepB variants containing the 19 amino acid segment that Ishii et al chose as the target of alanine scanning mutagenesis or the 26 amino acids segment having 9 additional N-terminally neighboring residues, but these fusion proteins proved to be incapable of inducing elongation arrest to the full extent in vivo. Therefore, we used the full-length mature sequence of VemP (delta-signal VemP) for EM analysis".

2. What does ST stand for in Figure 1? If it means stalled fragment, it does not apply to the Q156Stop constructs, which produce prematurely terminated, but not stalled, products. Incidentally, the letters indicating the constructs are too small to read.

3. The stalling efficiency is very low in the Figure 1 experiment, whereas it is almost 100% in the Figure 1 experiment. Why? Does hydrolysis of peptidyl-tRNA linkage take place under the conditions of the Figure 1 experiment? From the description in Material and Method, it seems that the authors used the former conditions for preparation of the cryo-EM samples. If so, the experimental system was accompanied by incomplete stalling or hydrolytic removal of the tRNA. Please clarify these issues.

---

## [Author Response]

*Reviewer #1:*

*1) The term "translocation" used in the manuscript refers to protein translocation, yet to someone in the translation (not secretion) field, this refers to the reaction catalyzed by EFG. This could be clarified throughout the manuscript.*

To be more precise, we replaced the term “translocation” with “protein translocation” throughout the manuscript (Introduction section and in the Legend for Figure 6).

*2) Discussion of Na^+^ regulation was a bit hard to integrate into the results of the manuscript. It might be worth clarifying that the results presented do not shed light on this particular subject but that this was how the motif was originally identified/studied.*

To better deliver the message that “VemP-dependent ribosome stalling is regulated by mechanic force” instead of “regulated by sodium concentration”, we have added a summary sentence in the Introduction, right after discussing the Na^+^ and Na^-^ conditions:

Introduction: “In brief, the mechanisms by which VemP switches protein synthesis between the SecDF1 and SecDF2 paralogs is dependent on the application of force to the signal sequence-containing VemP nascent chain. This force serves as a direct indicator of the SecDF-SecYEG protein translocation activity, which in turn is dependent on the environmental conditions.”

*3) More detailed comparison of the CMV stalling peptide and VemP could be made. While Figure 3 clearly shows that there is considerably more compaction in VemP peptide than in MifM or SecM, the distinction relative to the CMV peptide is less obvious. This should also be addressed in the text.*

To further compare the CMV stalling peptide with VemP, we have added one additional panel G to Figure 3 as well as two more panels, D and H, in Figure 5.

In Figure 3, we turned the view of the superposition of CMV and VemP about 180° and zoomed in to the C-terminal helical part, in order to show the difference in detail.

In Figure 5, the conformation of the analogous human U4493 and U4414 (*E. coli* U2585 and U2506) in the hCMV-SRC were compared to U2585 and U2506 in the VemP-SRC.

Legends for these figures have been revised accordingly. Moreover, extra descriptions have been added in the main text.

*4) The induced fit of the ribosome PTC was documented structurally by Schmeing et al., 2005 and the results here nicely agree. However, earlier biochemical work (Youngman et al., 2004) detailed an induced fit mechanism based on differences in reactivity of intact tRNAs and minimal A site substrates (like puromycin). It might be worth considering whether this study should also be cited.*

Thank you very much for reminding us of the earlier biochemical work from Youngman et al., We have cited this literature in our manuscript where induced fit of U2585 and U2506 were described (subsection “Silencing of the PTC by the C-terminal α-helix of VemP”).

*Reviewer #3:*

*1. Initially, I did not understand exactly what were the VemP constructs that the authors referred to "short" and "long". Inspection of Materials and Methods then made me realize that they were not really VemP constructs; rather, the authors inserted segments of VemP comprising the 19 and the 26 amino acids into the LepB sequence. The authors must have expected that these amino acid sequences would fully halt translational elongation when transplanted into a sequence context of an unrelated protein, LepB in this case. However, Ishii et al. only determined the C-terminal end of the VemP arrest sequence, leaving the N-terminal boundary obscure. Moreover, it is known that the arrest proficiency can be affected by the nature of the N-terminally adjacent sequences, although the LepB sequence may be expected to be inert in this respect. The results in Figure 1 show that neither the short nor the long construct exhibited full elongation arrest in vivo; less than 10% and ~50% efficiency, respectively, for "short" and "long". By contrast, the full-length delta-signal VemP gives nearly 100% arrest in vivo as shown by Ishii et al and in vitro as shown in Figure 1. In my opinion, the information content of Figure 1 is quite limited. Most importantly, the authors anyway used the full-length delta-signal VemP to prepare the samples of ribosome-nascent chain complexes for cryo-EM analysis. These considerations lead me to suggest that the clarity of the paper will be improved if the authors remove Figure 1 as well as the schemes of the short, the long and the GS constructs, from the main body of the manuscript. A simple description something like the following might be sufficient and appropriate: "We constructed LepB variants containing the 19 amino acid segment that Ishii et al chose as the target of alanine scanning mutagenesis or the 26 amino acids segment having 9 additional N-terminally neighboring residues, but these fusion proteins proved to be incapable of inducing elongation arrest to the full extent in vivo. Therefore, we used the full-length mature sequence of VemP (delta-signal VemP) for EM analysis".*

After discussion with co-authors, we decided we would prefer to keep Figure 1 in the main text. The reason is that this initial analysis of the VemP sequence on stalling efficiency resulted in the important conclusion that 7 or possibly even more Nterminal residues contribute to the stalling efficiency, in addition to the originally identified VemP stalling window. This is in agreement with our subsequent observation from the cryo-EM analysis that interactions between VemP and the ribosomal tunnel wall exist beyond the initial stalling window and even beyond the 7 residues. Moreover, showing this assay also explains why we decided to use an almost complete VemP sequence and not only the originally identified stretch for cryoEM as one could have expected.

To improve the clarity of the manuscript, we have revised and integrated your

suggestion to the text. The major revisions are the following:

Results, section “Analysis of VemP sequence on efficiency of stalling”:

“To ensure the stalling efficiency of the VemP constructs before initiating the structural analysis, we engineered three LepB-based constructs that carried C-terminal VemP residues, termed VemP “short”, “long” and “GS”, respectively (Figure 1). […] In contrast, VemP “long” containing the extended stalling window dramatically increased the stalling efficiency (Figure 1).”

and

“Taken together, these fusion proteins proved to be ineffective at inducing elongation arrest to the full extent in vivo.”

*2. What does ST stand for in Figure 1? If it means stalled fragment, it does not apply to the Q156Stop constructs, which produce prematurely terminated, but not stalled, products. Incidentally, the letters indicating the constructs are too small to read.*

Thank you for pointing this un-clarity from our Figure 1. For a more accurate labelling, we have marked each stalled fragment a white dot, and each full-length peptide including the controls a grey dot. The abbreviations were replaced by “full-length peptide” and “stalled peptide”, respectively. The font size for the letters indicating the constructs were enlarged to be comparable to other letters in the figure.

*3. The stalling efficiency is very low in the Figure 1 experiment, whereas it is almost 100% in the Figure 1 experiment. Why? Does hydrolysis of peptidyl-tRNA linkage take place under the conditions of the Figure 1 experiment? From the description in Material and Method, it seems that the authors used the former conditions for preparation of the cryo-EM samples. If so, the experimental system was accompanied by incomplete stalling or hydrolytic removal of the tRNA. Please clarify these issues.*

The major cause for the difference between Figure 1 was the source of the NuPAGE gels. For Figure 1, the NuPAGE gel was homemade and for Figure 1 a precast commercial NuPAGE gel. The appearance of the ratio of the peptidyl-tRNA to the free peptide was indeed lower in gel 1E, however, only due to an apparent pH deviation that hydrolyses to a larger extend the peptidyl-tRNA linkage when using the homemade gels. Therefore, as suggested by the referee, the different ratios indeed represent different degrees of peptidyl-tRNA hydrolysis in the different gel systems and not differences in the stalling efficiency. Moreover, single-particle image processing confirmed that the occupancy of proper P-site tRNA-bound ribosomes with high resolution was 80%, indicating good quality of sample preparation.